# Ranking by Relevance and Citation Counts, a Comparative Study: Google Scholar, Microsoft Academic, WoS and Scopus

**Cristòfol Rovira \***, **Lluís Codina**, **Frederic Guerrero-Solé and Carlos Lopezosa**

Department of Communication, Universitat Pompeu Fabra, 08002 Barcelona, Spain; lluis.codina@upf.edu (L.C.); frederic.guerrero@upf.edu (F.G.-S.); carlos.lopezosa@upf.edu (C.L.)
**\*** Correspondence: cristofol.rovira@upf.edu; Tel.: +34-667295308

**Abstract:** Search engine optimization (SEO) constitutes the set of methods designed to increase the visibility of, and the number of visits to, a web page by means of its ranking on the search engine results pages. Recently, SEO has also been applied to academic databases and search engines, in a trend that is in constant growth. This new approach, known as academic SEO (ASEO), has generated a field of study with considerable future growth potential due to the impact of open science. The study reported here forms part of this new field of analysis. The ranking of results is a key aspect in any information system since it determines the way in which these results are presented to the user. The aim of this study is to analyze and compare the relevance ranking algorithms employed by various academic platforms to identify the importance of citations received in their algorithms. Specifically, we analyze two search engines and two bibliographic databases: Google Scholar and Microsoft Academic, on the one hand, and Web of Science and Scopus, on the other. A reverse engineering methodology is employed based on the statistical analysis of Spearman's correlation coefficients. The results indicate that the ranking algorithms used by Google Scholar and Microsoft are the two that are most heavily influenced by citations received. Indeed, citation counts are clearly the main SEO factor in these academic search engines. An unexpected finding is that, at certain points in time, Web of Science (WoS) used citations received as a key ranking factor, despite the fact that WoS support documents claim this factor does not intervene.

**Keywords:** ASEO; SEO; reverse engineering; citations; google scholar; microsoft academic; web of science; WoS; scopus; indicators; algorithms; relevance ranking; citation databases; academic search engines

## 1. Introduction

The ranking of search results is one of the main challenges faced by the field of information retrieval [1,2]. Search results are sorted so that the results best able to solve the user's need for information are ranked at the top of the page [3]. The challenges faced though are far from straightforward given that a successful ranking by relevance depends on the correct analysis and weighting of a document's properties, as well as the analysis of the need for that information and the key words used [1,2,4].

Relevance ranking has been successfully employed in a number of areas, including web page search engines, academic search engines, academic author rankings and the ranking of opinion leaders on social platforms [5]. Many algorithms have been proposed to automate this relevance and some of them have been successfully implemented. In so doing, different criteria are applied depending on the specific characteristics of the elements to be ordered. PageRank [6] and Hyperlink-Induced Topic Search

(HITS) [7] are the best know algorithms for ranking web pages. Variants of these algorithms have also been used to rank influencers in social media, and include, for example, IP-Influence [8], TunkRank [9], TwitterRank [10] and TURank [11]. To search for academic documents, various algorithms have been proposed and used, both for the documents themselves and for their authors. These include Authority-Based Ranking [12], PopRank [13], Browsing-Based Model [14] and CiteRank [15]. All of them use the number of citations received by the articles as a search ranking factor in combination with other elements, such as publication date, the author's reputation and the network of relationships between documents, authors and affiliated institutions.

Many information retrieval systems (search engines, bibliographic databases and citation databases, etc.) use relevance ranking in conjunction with other types of sorting, including chronological, alphabetical by author, number of queries and number of citations. In search engines like Google, relevance ranking is the predominant approach and is calculated by considering more than 200 factors [16,17]. Unfortunately, Google does not release precise details about these factors, it only publishes fairly sketchy, general information. For example, the company says that inbound links and content quality are important [18,19]. Google justifies this lack of transparency in order to fight search engine spam [20] and to prevent low quality documents from being ranked at the top of the results by falsifying their characteristics.

Search engine optimization (SEO) is the discipline responsible for optimizing websites and their content to ensure they are ranked at the top of the search engine results pages (SERPs), in accordance with the relevance ranking algorithm [21]. In recent years, SEO has also been applied to academic search engines, such as Google Scholar and Microsoft Academic. This new application has received the name of "academic SEO" (or ASEO) [22–26]. ASEO helps authors and publishers to improve the visibility of their publications, thus increasing the chances that their work will be read and cited.

However, it should be stressed that the relevance ranking algorithm of academic search engines differs from that of standard search engines. The ranking factors employed by the respective search engine types are not the same and, therefore, many of those used by SEO are not applicable to ASEO while some are specific to ASEO (see Table 1).

SEO companies [27–29] routinely conduct reverse engineering research to measure the impact of the factors involved in Google's relevance ranking. Based on the characteristics of the pages that appear at the top of the SERPs, the factors with the greatest influence on the relevance ranking algorithm can be deduced. It is not a straightforward task since many factors have an influence and, moreover, the algorithm is subject to constant changes [30].

Studies that have applied a reverse engineering methodology to Google Scholar have shown that citation counts are one of the key factors in relevance ranking [31–34]. Microsoft Academic, on the other hand, has received less attention from the scientific community [35–38] and there are no specific studies of the quality of its relevance ranking.

Academic search engines, such as Google Scholar and Microsoft Academic, are an alternative to bibliographic commercial databases, such as Web of Science (WoS) and Scopus, for indexing scientific citations and they provide a free service of similar performance that competes with the business model developed by the classic services. Unlike search engines, bibliographic databases are fully transparent about how they calculate relevance, clearly informing users how their algorithm works on their help pages [39,40].

The primary aim of this study is to verify the importance attached to citations received in the relevance ranking algorithms of two academic search engines and two bibliographic databases. We analyze the two main academic search engines (i.e., Google Scholar and Microsoft Academic) and the two bibliographic databases of citations providing the most comprehensive coverage (WoS and Scopus) [41].

We address the following research questions: Is the number of citations received a key factor in Google Scholar relevance rankings? Do the Microsoft Academic, WoS and Scopus relevance algorithms operate in the same way as Google Scholar's? Do citations received have a similarly strong influence on all these systems? A similar approach to the one adopted here has been taken in previous studies of the factors involved in the ranking of scholarly literature [22,23,31–34].

The rest of this manuscript is organized as follows. First, we review previous studies of the systems that concern us here, above all those that focus on ranking algorithms. Next, we explain the research methodology and the statistical treatment performed. We then report, analyze and discuss the results obtained before concluding with a consideration of the repercussions of these results and possible new avenues of research.

## 2. Related Studies

Google Scholar, Microsoft Academic, WoS and Scopus have been analyzed previously in works that have adopted a variety of approaches, including, most significantly:

-   Comparative analyses of the coverage and quality of the academic search engines and bibliographic databases [42–51]
-   Studies of the impact of authors and the h-index [33,44,52–57]
-   Studies of the utility of Google Scholar and Academic Search for bibliometric studies [20,49,55,58–61]

However, few studies [43,62] have focused their attention on information retrieval and the search efficiency of academic search engines, while even fewer papers [22,23,31–34] have examined the factors used in ranking algorithms.

The main conclusions to be drawn from existing studies of relevance ranking in the systems studied can be summarized as follows:

-   The number of citations received is a very important factor in Google Scholar relevance rankings, so that documents with a high number of citations received tend to be ranked first [32–34].
-   Documents with many citations received have more readers and more citations and, in this way, consolidate their top position [61].

Surprisingly, the relevance ranking factors of academic search engines and bibliographic databases have attracted little interest in the scientific community, especially if we consider that a better position in their rankings means enhanced possibilities of being found and, hence, of being read. Indeed, the initial items on a SERP have been shown to receive more attention from users than that received by items lower down the page [63].

**Table 1.** Search engine optimization (SEO) and academic search engine optimization (ASEO) factors. WoS, Web of Science.

| Type | SEO/ASEO factor | Google Search | Google Scholar | Microsoft Academic | WoS | Scopus |
|---|---|---|---|---|---|---|
| On-page factors | Keywords in title | Yes [16–19,28–30] | Yes [31,32] | ? | Yes [40] | Yes [41] |
| | Keywords in URL, h1 or first words | Yes [16–19,28–30] | ? | ? | No [40] | No [41] |
| | Keyword frequency | No [16–19] | ? | ? | Yes [40] | Yes [41] |
| | Technical factors: design, speed, etc. | Yes [16–19,28–30] | ? | ? | No [40] | No [41] |
| Off-page factors | Backlinks | Yes [16–19,28–30] | ? | ? | No [40] | No [41] |
| | Received citations | ? | Yes [16,31–34] | Yes [35–38,64] | No [40] | No [41] |
| | Author reputation | Yes [16–19,28–30] | Yes [16] | Yes [35–38,64] | No [40] | No [41] |
| | Reputation of the publication or domain | Yes [16–19,28–30] | Yes [16] | Yes [35–38,64] | No [40] | No [41] |
| | Signals from social networks | Yes (Indirect) [16–19] | ? | ? | No [40] | No [41] |
| | Traffic, Click Through Rate | Yes [16–19,28–30] | ? | ? | No [40] | No [41] |
| Artificial intelligence | RankBrain | Yes [17–19,28–30] | ? | ? | No [40] | No [41] |

In the light of these previous reports, it can be concluded that the number of intervening factors in the academic search engines is likely to be fewer than those employed by Google and that, therefore, the algorithm is simpler (see Table 1).

## 3. Methodology

This study is concerned with analyzing the relevance ranking algorithms used by academic information retrieval systems. We are particularly interested in identifying the factors they employ, especially in the case of systems that do not explain how their ranking algorithm works. A reverse engineering methodology is applied to the two academic search engines (i.e., Google Scholar and Microsoft Academic) and to two bibliographic databases of citations (i.e., WoS and Scopus). These, in both cases, are the systems offering the most comprehensive coverage [41,65,66]. The specific objective is to identify whether the citations received by the documents are a determining factor in the ranking of search results.

Reverse engineering is a research method commonly used to study any type of device in order to identify how it works and what its components are. It is a relatively economical way to obtain information about the design of a device or the source code of a computer program based on the compiled files.

One of the fields in which reverse engineering is being applied most is precisely in that of detecting the factors included in Google's relevance ranking algorithm [28,29,67]. The little information provided by Google [16] is used as a starting point to analyze the characteristics of the pages ranked at the top of the search results to deduce what factors are included and what their respective weighting is. However, the ranking algorithms are complex [68], moreover, they are subject to frequent modifications and the results of reverse engineering are usually inconclusive. Recently, a reverse engineering methodology has also been applied to academic search engines [34].

To obtain an indication of the presence of a certain positioning factor, the ranking data are treated statistically by applying Spearman's correlation coefficient, selected here because the distribution is not normal according to Kolmogorov–Smirnov test results. Generally, a comparison is made of the ranking created by the researcher using the values of the factor under study with the search engine's natural ranking—for example, a ranking based on the frequency of the appearance of the keywords in the document and Google's native ranking. If a high coefficient is obtained, this means that this factor is contributing significantly to the ranking. However, in the case of Google, many factors intervene: more than 200, according to many sources [69,70]. Therefore, it is very difficult to detect high correlations indicative of the fact that a certain characteristic has an important weighting. Statistical studies generally consider a correlation between 0.4 and 0.7 to be moderate and a correlation above 0.7

to be high. In reverse engineering studies with Google, the correlation values between the positions of the pages and the quantitative values of the supposed positioning factors do not normally exceed 0.3 [68]. Although the correlations are low, with studies of this type, relatively clear indications of the factors intervening in the ranking can be obtained.

Google themselves provide even less data on how they rank by relevance in Google Scholar. Perhaps their most explicit statement is the following:

> "Google Scholar aims to rank documents the way researchers do, weighing the full text of each document, where it was published, who it was written by as well as how often and how recently it has been cited in other scholarly literature." [71].

Previous research [64,72] has shown that Google Scholar applies far fewer ranking factors than is the case with Google's general search engine. This is a great advantage when applying reverse engineering since the statistical results are much clearer, with some correlations being as high as 0.9 [34].

Likewise, Microsoft Academic does not offer any specific details about its relevance ranking algorithm [73]. We do know, however, that it applies the Microsoft Academic Graph or MAG [74], an enormous knowledge database made up of interconnected entities and objects. A vector model is applied to identify the documents with the greatest impact using the PopRank algorithm [13,75]. However, Microsoft Academic does not indicate exactly what the "impact" is when this concept is applied to the sorting algorithm:

> "In a nutshell, we use the dynamic eigencentrality measure of the heterogeneous MAG to determine the ranking of publications. The framework ensures that a publication will be ranked high if it impacts highly ranked publications, is authored by highly ranked scholars from prestigious institutions, or is published in a highly regarded venue in highly competitive fields. Mathematically speaking, the eigencentrality measure can be viewed as the likelihood that a publication will be mentioned as highly impactful when a survey is posed to the entire scholarly community" [76]

Unlike these two engines, the WoS and Scopus bibliographic databases provide detailed information about their relevance ranking factors [39,40]. In systems of this type, a vector model is applied [1] and relevance is calculated based on the frequency and position of the keywords of the searches in the documents; therefore, citations received are not a factor.

Another factor that facilitates the use of reverse engineering in the cases of Google Scholar, Microsoft Academic, WOS and Scopus is the information these systems provide regarding the exact number of citations received, a factor used to compute their rankings. Unlike the general Google search engine that does not give reliable information about the number of inbound links, in the four systems studied the number of citations received, and even the listing of all citing documents, is easily obtained. The relative simplicity of the algorithms and the accuracy of the citation counts mean reverse engineering is especially productive when applied to the study of the influence of citations in relevance ranking in academic search engines and bibliographic databases of citations.

For the study reported here, 25,000 searches were conducted in each system, a similar number to those typically conducted in reverse engineering studies [27–29] or other analyses of Google Scholar rankings [22,23,31–34]. The ranking provided by each tool was then compared with a second ranking created applying only the number of citations received. As the distributions were not normal according to Kolmogorov–Smirnov test results, Spearman's correlation coefficient was calculated. The hypothesis underpinning reverse engineering as applied to search engine results is that the higher the correlation coefficient, the more similar the two rankings are and, therefore, a greater weight can be attributed to the isolated factor used in the second ranking.

To avoid thematic biases, the keywords selected for use in the searches needed to be as neutral as possible. Thus, we chose the 25 most frequent keywords appearing in academic documents [77–79].

Searches were then conducted in Google Scholar and Microsoft Academic using these keywords which also enabled us to identify two-term searches based on the suggestions made by these two engines. Next, from among these suggestions we selected those with the greatest number of results. In this way, we obtained two sets of 25 keywords, the first formed by a single term and the second by two. The terms used can be consulted in Annex 1. It is critical that each search provides as many results as possible in order to ensure that the statistical treatment of the rankings of these results is robust. It is for this reason that we didn't use Long Tail Keywords.

To address our research question concerning the impact of citations received on ranking, we undertook searches with these keywords in each system collecting up to 1000 results each time. In the case of the academic search engines, searches were carried out with both one and two-term keywords. In the case of the bibliographic databases, searches were only carried out with two-term keywords since our forecasts were very clear in indicating that citations did not affect the results—indeed, the documentation for these systems also make this quite clear. However, as we see below, the results were not as expected.

The search engine data were obtained using the Publish or Perish tool [80,81] between 10 May 2019 and 30 May 2019 (see Appendices A and B). The data from the bibliographic databases were obtained by exportation from the systems themselves between these same dates.

In each of the systems studied, our rankings created using citation counts were compared with the native rankings of each system. To do this, the number of citations received was transformed to an ordinal scale, a procedure previously used in other studies [31,32]. According to reverse engineering methodology, if the two rankings correlate then they are similar and, therefore, it can be deduced that citations are an important factor in the relevance ranking algorithms. The ranking by citations received was correlated for each of the 25 searches (and their corresponding 1000 results) carried out in each system with the native ranking of these systems.

To obtain a global value for each system which integrates the 25 searches and their corresponding 25,000 data items, the median values of each of the 25 citation search rankings were used for each position in the native ranking. The native ranking positions of each system are shown on the x-axis, while the rankings according to citations received are shown on the y-axis. Each gray dot corresponds to one of the 25,000 data items from the 25 searches of 1000 results each conducted on each system. The blue dots are the 1000 median values that indicate the central tendency of the data. The more compact and the closer to the diagonal the medians are, the greater the correlation between the two rankings.

The software used in the analysis was R, version 3.4.0 [82] and SPSS, version 20.0.0. The confidence intervals were constructed via normal approximation by applying Fisher's transformation using the R psych package [83,84]. Fisher's transformation when applied to Spearman's correlation coefficient is asymptotically normal. Graphs were drawn with Google Sheets and Tableau.

## 4. Analysis of Results

Table 2 shows the results obtained when analyzing the four systems. It can be seen that in some cases different analyses were conducted on the same system. This reflects various circumstances impacting the study. For example, in the case of Microsoft Academic we did not perform a full-text search, rather it was limited to the bibliographic reference: that is, the title, keywords, name of publication, author and the abstract. Interestingly, in conducting the study we found that in more than 95% of the searches the keywords were present in the result titles. This gave rise to a problem when we compared the results with Google Scholar, since this search engine does perform full-text searches. For this reason, we undertook a second data collection in the case of Google Scholar, restricting searches to the title and, in this way, we are able to make a more accurate comparison of the results provided by Google Scholar and Microsoft Academic. The introduction of a second variant was a result of the number of search terms. The study was carried out using two sets of 25 keywords, the first made up of a single term and the second of two terms. Finally, in the case of WoS, two data collections were

undertaken since it became apparent that the ranking criteria had changed. However, each of these variants allows us to make partial comparisons and to analyze specific aspects of the systems.

**Table 2.** Correlation coefficients for the academic search engines.

| System | Number of Search Terms | Search Restrictions Included | Spearman's Coefficient | $p$ |
|---|---|---|---|---|
| Google Scholar | 1 | unrestricted | 0.968 | <0.0001 |
| Google Scholar | 2 | unrestricted | 0.721 | <0.0001 |
| Google Scholar | 1 | title | 0.990 | <0.0001 |
| Google Scholar | 2 | title | 0.994 | <0.0001 |
| Microsoft Academic | 1 | title, abstract, keywords | 0.907 | <0.0001 |
| Microsoft Academic | 2 | title, abstract, keywords | 0.937 | <0.0001 |
| Scopus | 2 | title, abstract, keywords | −0.107 | <0.001 |
| WoS-version 1 | 2 | title, abstract, keywords | −0.075 | <0.05 |
| Wos-version 2 | 2 | title, abstract, keywords | 0.907 | <0.0001 |

The correlations for the two academic search engines were, in all cases, higher than 0.7 and in some cases reached 0.99 (Table 2). These results indicate that citation counts are likely to constitute a key ranking factor with considerable weight in the sorting algorithm. The other factors can cause certain results to climb or fall down the ranking, but the main factor would appear to be citations received. In the case of the bibliographic databases, the correlations were close to zero and, therefore, in such systems, we have no evidence that citations intervene. However, in the case of WoS, over a period of several days, we detected it to be using different sorting criteria and its results were in fact ranked using the number of citations received as its primary factor. This result is surprising since it does not correspond to the criteria the database claims to use in its corresponding documentation. We attribute these variations to tests being performed to observe user reactions, and on the basis of these results, decisions can presumably be taken to modify the ranking algorithm, but this is no more than an inference.

We find virtually no differences between searches conducted with either one or two terms in the academic search engines (see Figures 1–6). In principle, there is no reason as to why there should be any differences as the same ranking algorithm was applied in all cases. However, we did find some differences in the case of Google Scholar when searches were performed with or without restriction to the title, the same search providing a different ranking. When the terms were in all the titles, correlation coefficients of 0.9 were obtained (see Figures 3–6). When this was not the case, the correlation coefficient was still very high, but fell to 0.7 in the searches with two words (see Figure 2). This difference of almost 20 points is a clear indication that the inclusion of keywords in the title is also an important positioning factor. These differences are even more evident if we analyze the correlation coefficients of each search. Figure 7 shows how in all cases the correlation coefficients of the searches performed with this restriction are greater than when this restriction is not included. When restricting searches to the title, this factor is nullified since all the results have it. In such circumstances, the correlation is almost 1 since practically the only factor in operation is that of citations received. Therefore, indirectly, we can verify that the inclusion of the search terms in the document title forms part of the sorting algorithm, which we already knew, but for which we are now able to provide quantitative evidence showing that the weight of this factor is lower than that of citations since there is little difference between the two correlations. Unfortunately, this same analysis cannot be conducted in the case of Microsoft Academic because it does not permit full-text searches.

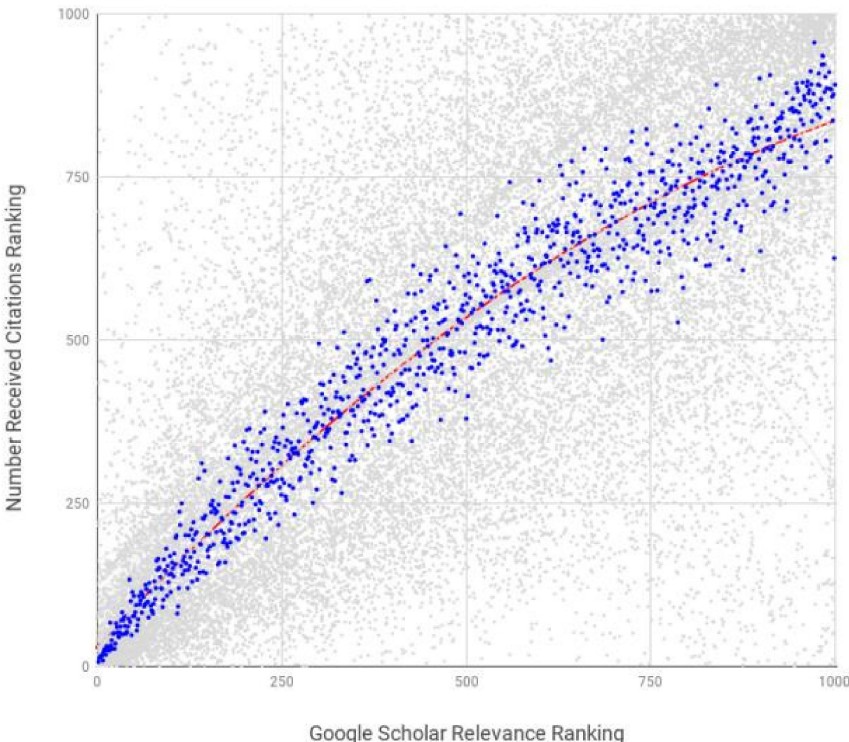

**Figure 1.** Google Scholar Searches with One Word and No Title Restriction (rho 0.968, median in blue, the rest of data in gray).

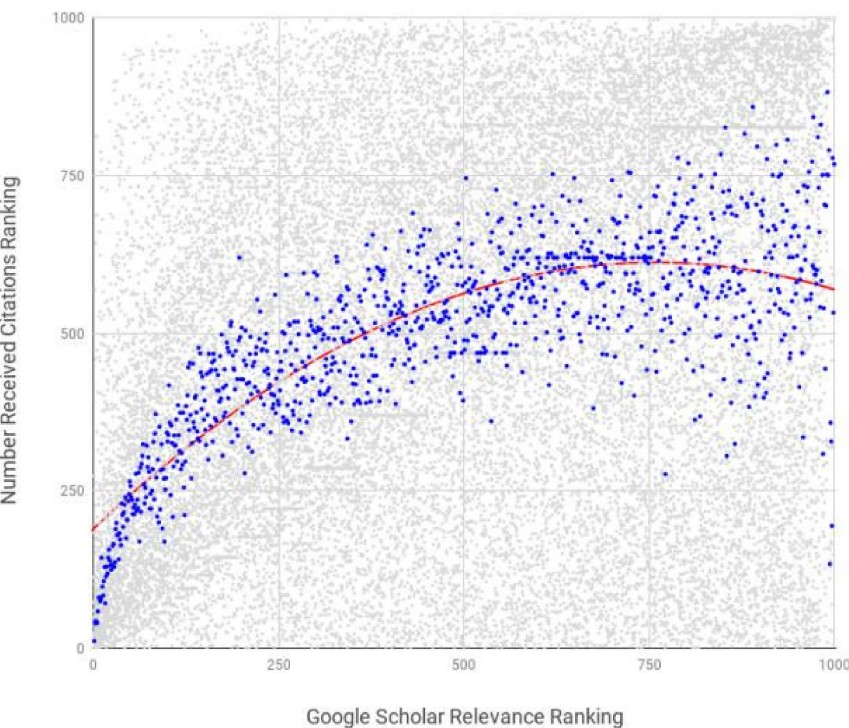

**Figure 2.** Google Scholar Searches with Two Words and No Title Restriction (rho 0.721, median in blue, the rest of data in gray).

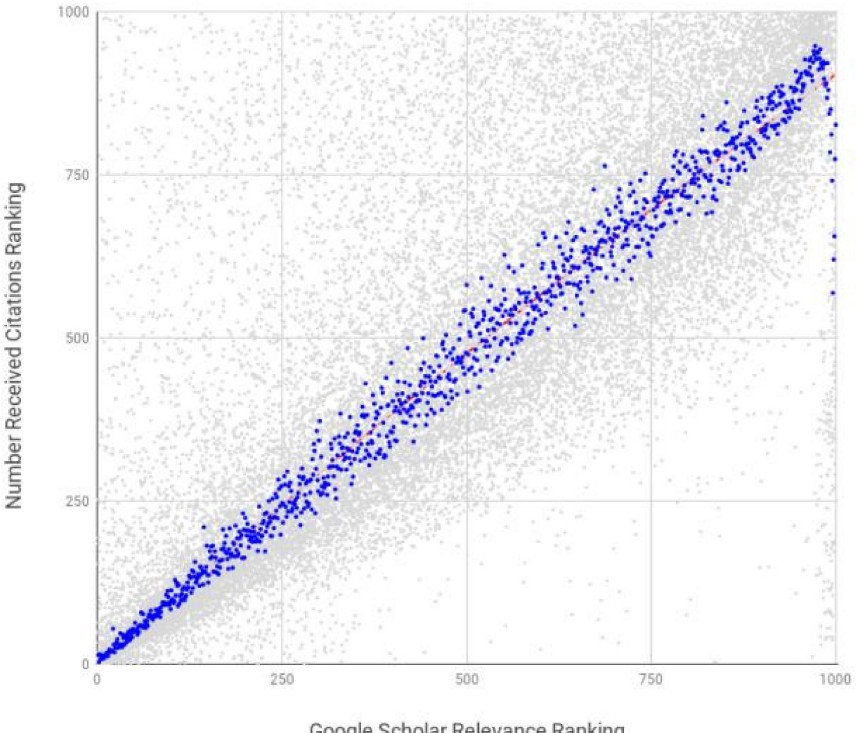

**Figure 3.** Google Scholar Searches with One Word and Title Restriction (rho 0.990, median in blue, the rest of data in gray).

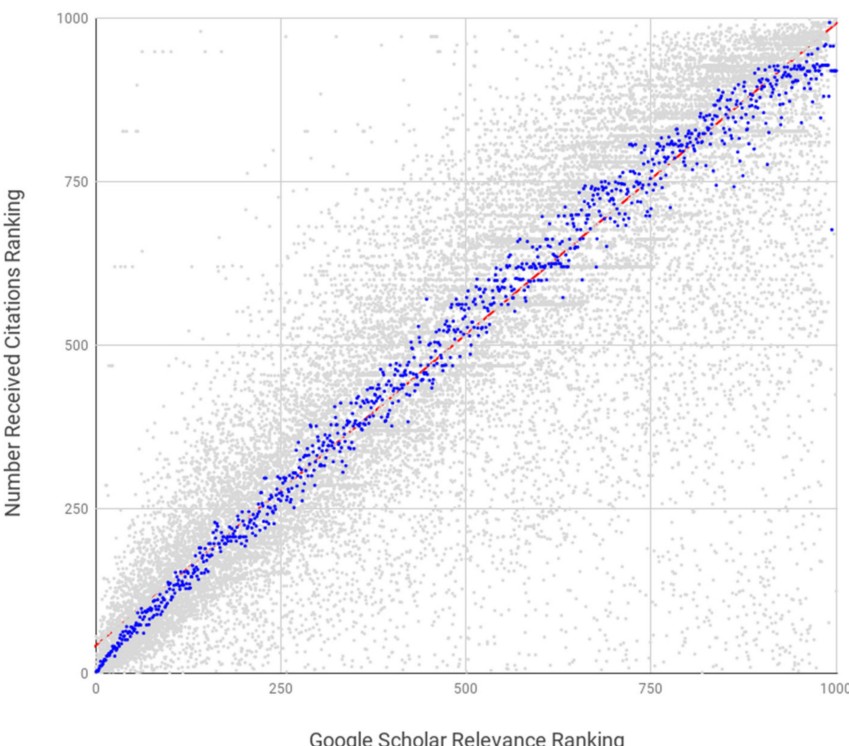

**Figure 4.** Google Scholar Searches with Two Words and Title Restriction (rho 0.994, median in blue, the rest of data in gray).

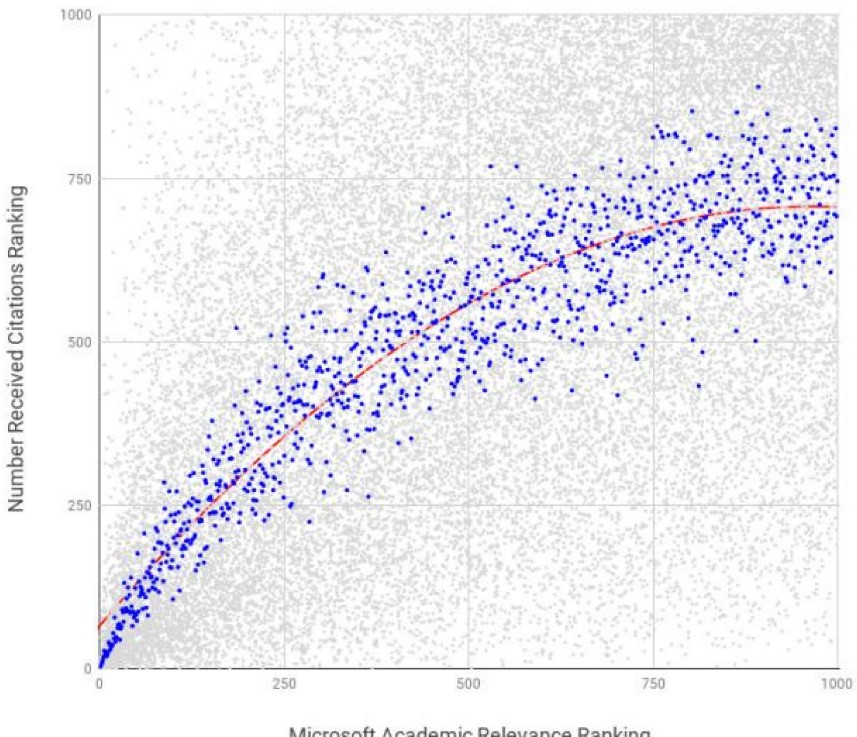

**Figure 5.** Microsoft Academic Searches with One Word and Title Restriction (rho 0.907, median in blue, the rest of data in gray).

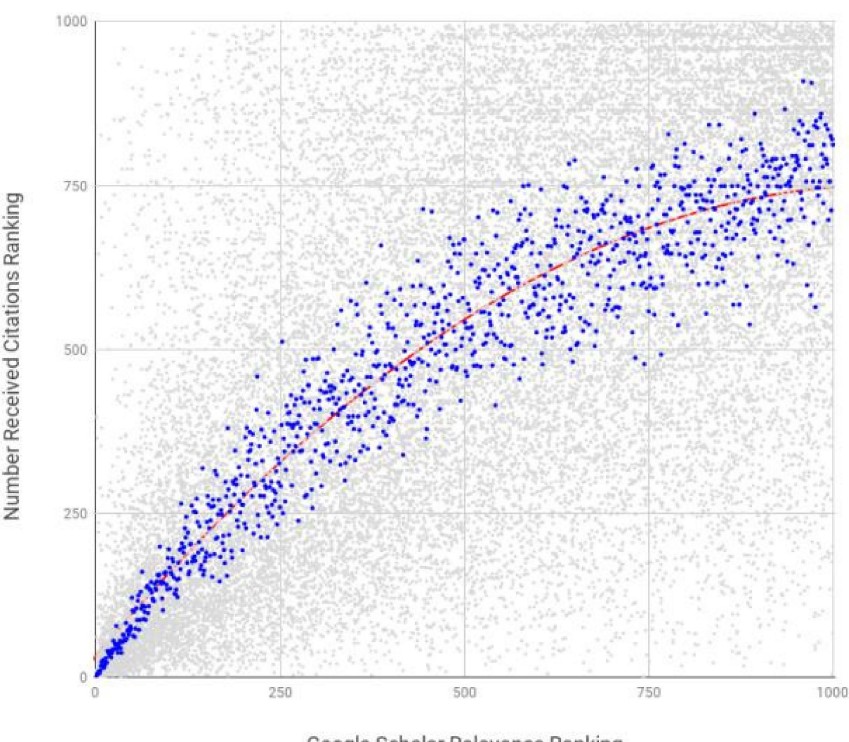

**Figure 6.** Microsoft Academic Searches with Two Words and Title Restriction (rho 0.937, median in blue, the rest of data in gray).

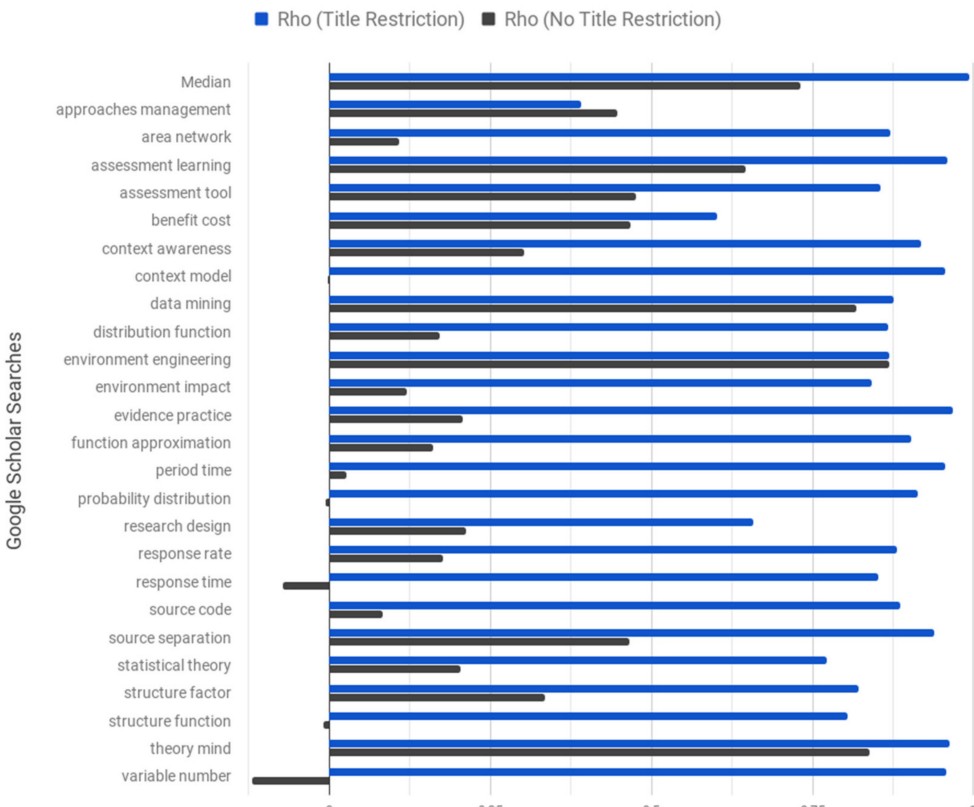

**Figure 7.** Google Scholar: Title Restriction vs No Title Restriction with Two Words.

Likewise, we detected no differences between Google Scholar and Microsoft Academic, above all when comparing the results of searches restricted to the title (Figures 3 and 6). In all four cases, we obtained correlation coefficients of 0.9. Therefore, it seems that both academic search engines apply a very similar weight to the factor of citations received.

Finally, it is worth noting that the two bibliographic databases (see Figures 8 and 9) do not employ citations received as a positioning factor, as stated in their documentation. Therefore, it is perfectly logical that their corresponding correlation coefficients are almost zero. However, somewhat surprisingly, in the case of WoS, we found that between 20 May 2019 and 25 May 2019 the ranking was different and we obtained a correlation coefficient similar to that of the academic search engines, i.e., 0.9 (see Figures 10–12) and that, therefore, a different algorithm was being applied with the significant inclusion of citation counts. It is common that before introducing changes in the design of websites, tests are made with real users. Figures 10 and 11 illustrate screenshots of the same search but employing two different relevance rankings. This can be done by randomly publishing different prototypes to gather information on user behavior in order to determine which prototype achieves greatest acceptance. As discussed above, it would seem that WoS was implementing such a procedure and was carrying out tests aimed at modifying its relevance ranking using an algorithm similar to that of academic search engines, although we should insist that this is only an inference.

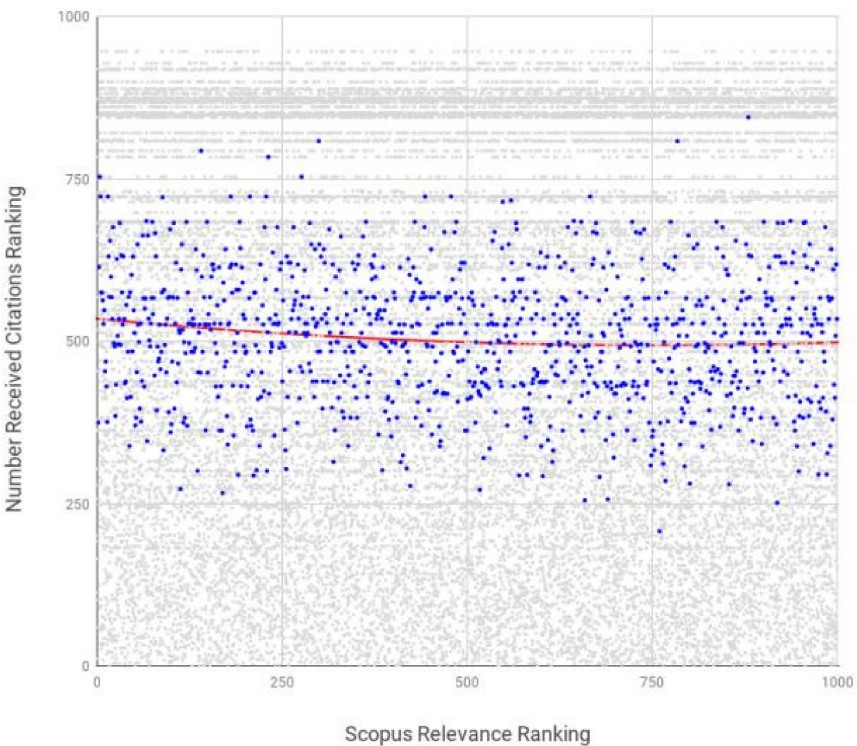

**Figure 8.** Scopus Searches with Two Words and Title Restriction (rho −0.10, median in blue, the rest of data in gray).

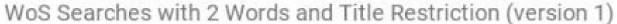
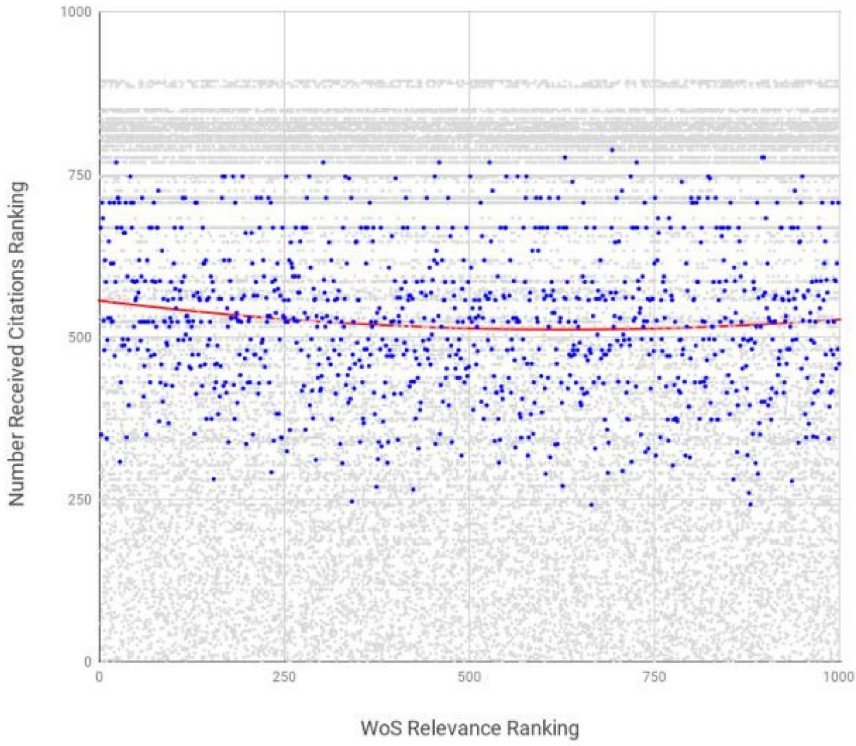

**Figure 9.** WoS Searches with Two Words and Title Restriction (Version 1) (rho −0.075, median in blue, the rest of data in gray).



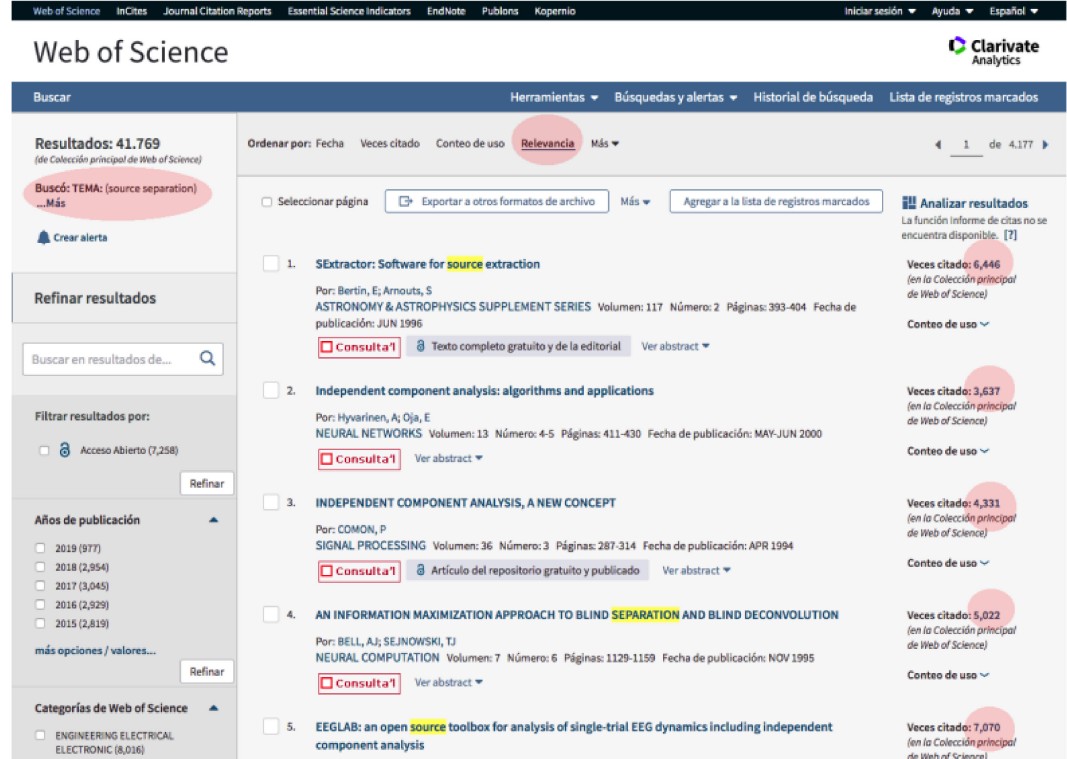

**Figure 10.** Search conducted on WoS with relevance ranking using the number of citations.

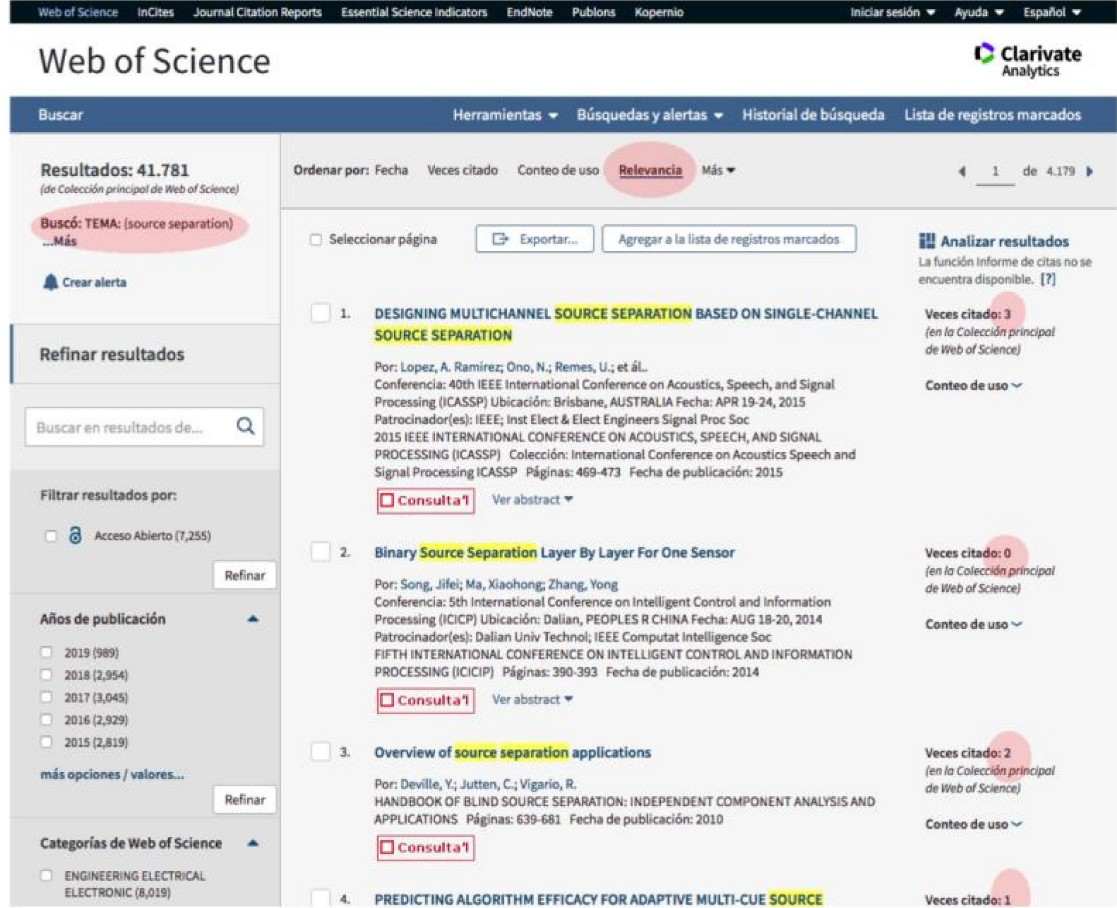

**Figure 11.** Same search as in Figure 10 with relevance ranking but without using the number of citations.

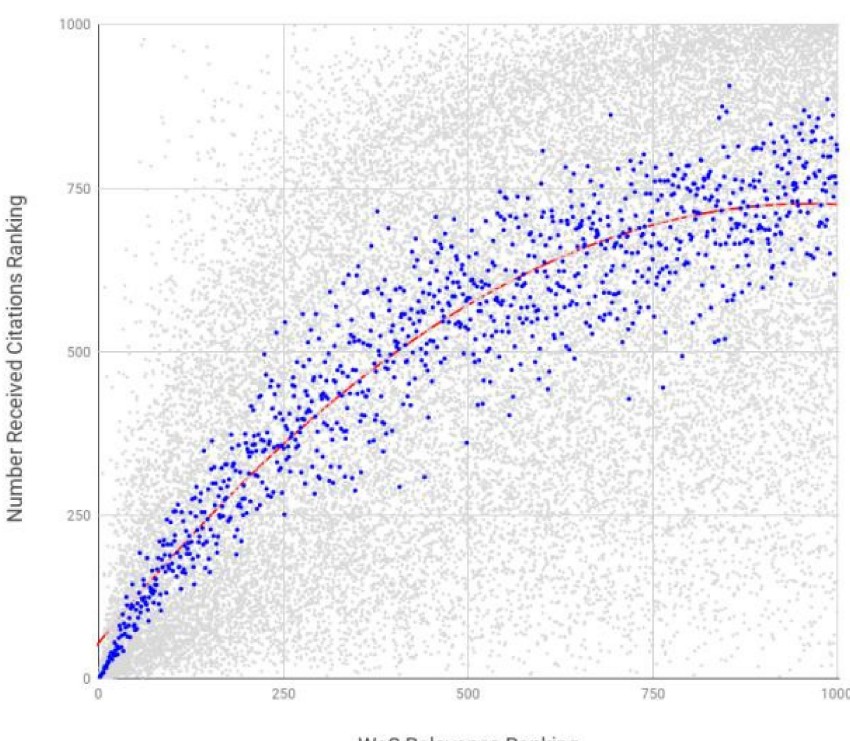

**Figure 12.** WoS Searches with Two Words and Title Restriction (Version 2) (rho 0.907, median in blue, the rest of data in gray).

## 5. Discussion

The importance attached to citations received in Google Scholar ranking of results of searches is not exactly a new finding. Beel and Gipp [31–34] described the great importance of this factor, both in full-text searches and searches by title only. However, our study incorporates methodological improvements on these earlier studies giving greater consistency to our results. Beel and Gipp applied a very basic statistical treatment, drawing conclusions from an analysis of scatter plots but without calculating correlation coefficients or conducting other specific statistical tests. Moreover, to obtain a global value of various rankings the authors took the mean. It is our understanding that the more appropriate measure of central tendency for ordinal variables is the median. Finally, the words used by the authors when conducting their searches were randomly selected from an initial list. This procedure generated a number of problems since many searches did not generate any results. In contrast, the procedure applied in the study described here is based on the most frequent words in academic documents [77–79] and the searches suggested by the academic search engines themselves based on an analysis of a large volume of user searches. This procedure ensures the random selection of the searched content and that in the vast majority of searches there are at least 1000 results. Future studies need to confirm that searches providing few results apply the same ranking criteria, as would be expected.

Beel and Gipp [31,32] found that citations received were more influential in full-text searches than they were in those restricted to the title only. Their conclusions were that Google Scholar applied two slightly different algorithms depending on the search type. Our results differ on this point as we detect a greater weighting for citations in searches restricted to just the title. There is no reason, however, to believe that different algorithms are being applied, rather it would appear to be a case of the same algorithm behaving differently depending on the factors that intervene. The presence of search words in the title is a positioning factor that forms part of the algorithm. If we ensure that all the results have the search words in the title, then we cancel out this factor and the effect of citations received is very

clear. On the other hand, in full-text searches this factor does intervene and, therefore, the influence of citations received is less clear since the ranking is also determined by the presence or otherwise of the words in the title.

In a study conducted by Martín-Martín et al. [33], the authors found that in Google Scholar the citations received also had a strong influence on searches by year of publication. The authors calculated Pearson's correlation coefficient and obtained values above 0.85. These results are similar to those obtained in our study. However, Martín-Martín et al. [33] adopted a somewhat unusual method for calculating the overall value of all the searches conducted, taking the arithmetic mean of the correlation coefficients. It is our understanding that to obtain a measure of central tendency shortcuts cannot be taken and it is more appropriate to obtain the median for each position and then calculate the correlation coefficient of these medians for the Google Scholar ranking.

In Rovira et al. [34], the authors focused their attention on the weight of citations received in the relevance ranking, but only in the case of Google Scholar. While in this earlier study the authors considered searches by year, author, publication and the "cited by" link, searches by keyword were not examined. However, a very similar conclusion was reached regarding citations received: namely, that they are a very important relevance ranking factor in Google Scholar. The present study has expanded this earlier work by analyzing other information retrieval systems using keyword searches, the most common search type conducted.

Finally, it is worth stressing that we have not found any previous reports on the specific criteria for the relevance ranking used by the other three systems analyzed here. As such, we believe our study provides new reliable data on these systems.

Relevance is a concept that is clearly very much open to interpretation since it seeks to identify items presenting the highest quality, a characteristic with a very strong subjective element. The diversity of algorithms for determining relevance is a clear indicator of the complexity of its automation. For this reason, citations received is granted so much weight.

## 6. Conclusions

Our results indicate that citation counts are probably the main factor employed by Google Scholar and Microsoft Academic in their ranking algorithms. In the case of Scopus, by contrast, we find no evidence that citations are taken into account, as indeed is reported in the database's supporting documentation [39].

In the specific case of WoS, we detected two distinct rankings. In the initial data collection exercise, the ranking of results was conducted according to the criteria described in the WoS documentation, that is, without applying citation counts and weighting the results according to the position and frequency of keywords, as Elsevier [40] states in its documentation for this service. However, somewhat surprisingly, in a second data gathering process it became evident that the ranking on this occasion was, in essence, based on citations received. It would seem that these two distinct ranking systems were detected because WoS was undertaking tests with a view to changing its algorithm and, as such, modified its ranking criteria to obtain a better understanding of user behavior.

Our findings allow us to improve the experimental foundations of ASEO and enable us to offer useful suggestions to authors as to how they might optimize the ranking of their research in the main academic information retrieval systems. Greater visibility is implicit of a greater probability of their being read and cited [61,63] and, thereby, of boosting authors' chances to improve their h-index [85]. Any information that allows us to identify the factors that intervene in relevance ranking is of great value, not so that we might manipulate the ranking results—something that is clearly undesirable—but rather so that we can take them into account when promoting the visibility of the academic production of an author or a research group.

Other academic databases are emerging, including Dimensions and Lens, but they do not provide the same coverage as the two databases considered here. Nevertheless, we cannot rule out the possibility of their being analyzed in future studies. Such studies should usefully seek to undertake

the simultaneous analysis of various factors, including, for example, citations received and keywords in a document's title, as we have discussed above. One of the limitations of this study is precisely that a single factor is studied in isolation, when a ranking algorithm employs many factors simultaneously. It would be of particular interest to analyze whether such algorithms employ interactions between several factors.

**Author Contributions:** Conceptualization, C.R.; Methodology, C.R.; Validation L.C., F.G-S. and C.L.; Investigation C.R., L.C. and F.G.-S.; Resources, C.R. and C.L.; Data Curation, C.R.; Writing—Original Draft Preparation, C.R.; Writing—Review and Editing, L.C., F.G.-S. and C.L.; Supervision L.C. and F.G.-S.

**Funding:** This research was funded by the project "Interactive storytelling and digital visibility in interactive documentary and structured journalism". RTI2018-095714-B-C21, ERDF and Ministry of Science, Innovation and Universities (Spain).

**Conflicts of Interest:** The authors declare no conflicts of interest.

## Appendix A. List of Terms Used in The Searches

### One-Term Searches
Search words obtained from [78].

**Table A1.** Words and Rho of one-term searches. ** $p < 0.01$.

| Search Words | Rho Google Scholar Title | Rho Google Scholar Not Title | Rho Microsoft Academic Title |
|:---:|:---:|:---:|:---:|
| Median | 0.990 ** | 0.968 ** | 0.907 ** |
| approach | 0.742 ** | 0.700 ** | 0.548 ** |
| assessment | 0.632 ** | 0.563 ** | 0.545 ** |
| authority | 0.868 ** | 0.815 ** | 0.465 ** |
| consistent | 0.956 ** | 0.783 ** | 0.596 ** |
| context | 0.851 ** | 0.681 ** | 0.483 ** |
| data | 0.645 ** | 0.662 ** | 0.601 ** |
| definition | 0.907 ** | 0.783 ** | 0.636 ** |
| derived | 0.905 ** | 0.682 ** | 0.568 ** |
| distribution | 0.781 ** | 0.649 ** | 0.458 ** |
| estimate | 0.939 ** | 0.813 ** | 0.527 ** |
| evidence | 0.761 ** | 0.616 ** | 0.488 ** |
| fact | 0.899 ** | 0.229 ** | 0.517 ** |
| factor | 0.490 ** | 0.591 ** | 0.490 ** |
| formula | 0.872 ** | 0.773 ** | 0.352 ** |
| function | 0.789 ** | 0.650 ** | 0.529 ** |
| interpretation | 0.852 ** | 0.723 ** | 0.570 ** |
| method | 0.762 ** | 0.665 ** | 0.613 ** |
| percent | 0.932 ** | 0.861 ** | 0.478 ** |
| principle | 0.879 ** | 0.812 ** | 0.530 ** |
| research | 0.500 ** | 0.642 ** | 0.521 ** |
| response | 0.741 ** | 0.603 ** | 0.500 ** |
| significant | 0.929 ** | 0.709 ** | 0.557 ** |
| source | 0.848 ** | 0.735 ** | 0.546 ** |
| theory | 0.488 ** | 0.544 ** | 0.483 ** |
| variable | 0.888 ** | 0.727 ** | 0.569 ** |

### Two-Term Searches
Search words obtained from the above list and by selecting the search suggestions provided by Google Scholar and Microsoft Academic with the greatest number of results.

**Table A2.** Words and Rho of two-term searches. ** $p < 0.01$, * $p < 0.05$.

| Search Words | Rho GS Title | Rho GS Not Title | Rho MA Title | Rho Scopus | Rho WoS Version 1 | Rho WoS Version 2 |
|---|---|---|---|---|---|---|
| Median | 0.994 ** | 0.721 ** | 0.937 ** | −0.107 ** | −0.075* | 0.907 ** |
| approaches management | 0.391 ** | 0.447 ** | 0.587 ** | −0.004 | −0.102 ** | 0.581 ** |
| area network | 0.871 ** | 0.108 ** | 0.462 ** | 0.025 | −0.054 | 0.610 ** |
| assessment learning | 0.960 ** | 0.646 ** | 0.683 ** | 0.009 | −0.038 | 0.605 ** |
| assessment tool | 0.855 ** | 0.476 ** | 0.619 ** | −0.006 | 0.058 | 0.556 ** |
| benefit cost | 0.602 ** | 0.467 ** | 0.490 ** | −0.048 | −0.008 | 0.522 ** |
| context awareness | 0.918 ** | 0.302 ** | 0.752 ** | −0.066* | −0.056 | 0.580 ** |
| context model | 0.956 ** | −0.003 | 0.616 ** | −0.042 | 0.023 | 0.624 ** |
| data mining | 0.875 ** | 0.818 ** | 0.747 ** | 0.072* | 0.009 | 0.654 ** |
| distribution function | 0.868 ** | 0.171 ** | 0.415 ** | −0.051 | −0.069* | 0.520 ** |
| environment engineering | 0.869 ** | 0.869 ** | 0.539 ** | −0.037 | −0.032 | 0.647 ** |
| environment impact | 0.842 ** | 0.120 ** | 0.559 ** | −0.026 | −0.065* | 0.605 ** |
| evidence practice | 0.968 ** | 0.206 ** | 0.712 ** | 0.021 | 0.005 | 0.517 ** |
| function approximation | 0.903 ** | 0.162 ** | 0.619 ** | 0.053 | −0.036 | 0.646 ** |
| period time | 0.956 ** | 0.027 | 0.525 ** | −0.102 ** | 0.036 | 0.554 ** |
| probability distribution | 0.913 ** | −0.006 | 0.522 ** | −0.077* | −0.055 | 0.683 ** |
| research design | 0.658 ** | 0.213 ** | 0.648 ** | 0.099 ** | 0.063* | 0.572 ** |
| response rate | 0.881 ** | 0.176 ** | 0.522 ** | −0.035 | 0.114 ** | 0.606 ** |
| response time | 0.851 ** | −0.072* | 0.469 ** | −0.016 | −0.043 | 0.543 ** |
| source code | 0.887 ** | 0.082 ** | 0.631 ** | −0.119 ** | 0.043 | 0.621 ** |
| source separation | 0.939 ** | 0.466 ** | 0.734 ** | 0.053 | −0.062 | 0.588 ** |
| statistical theory | 0.772 ** | 0.203 ** | 0.573 ** | −0.085 ** | −0.081* | 0.657 ** |
| structure factor | 0.821 ** | 0.334 ** | 0.575 ** | −0.209 ** | −0.026 | 0.607 ** |
| structure function | 0.804 ** | −0.01 | 0.582 ** | −0.208 ** | −0.104 ** | 0.613 ** |
| theory mind | 0.962 ** | 0.838 ** | 0.680 ** | −0.048 | 0.181 ** | 0.672 ** |
| variable number | 0.958 ** | −0.120 ** | 0.656 ** | 0.072* | −0.001 | 0.081* |

## Appendix B. Data Files

Rovira, C.; Codina, L.; Guerrero-Solé, F.; Lopezosa, C. Data set of the article: Ranking by relevance and citation counts, a comparative study: Google Scholar, Microsoft Academic, WoS and Scopus (Version 1) (Data set). Zenodo. Available online: http://doi.org/10.5281/zenodo.3381151 (accessed on 10 September 2019).

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
