# Peer review of "Ranking by Relevance and Citation Counts, a Comparative Study: Google Scholar, Microsoft Academic, WoS and Scopus"

_futureinternet, doi:10.3390/fi11090202_

Round 1

Reviewer 1 Report

The article is coherent and cohesive. The theme is relatively original and the methodology used is appropriate. However, two small aspects should be dealt with. On the one hand, it lacks a section of conclusions, so it would be interesting to add it. On the other hand, the results of the research are detailed at the end of the introduction. This information should not appear in the introduction.

Reviewer 2 Report

Dear authors,

Please find an annotated version of your paper attached. I've made a number of suggested edits to add clarity in certain places. I found this quite a good read, thank you. I think it will be of interest to quite a wide audience, and look forward to seeing it published.

Best,

Reviewer 3 Report

I am pleased to have the opportunity to review this research paper. This study attempted to contribute to the analysis of ranking by relevance and citation counts, in Google Scholar, Microsoft Academic, Was and Scopus. Although the research is interesting there are significant errors that should be corrected before consideration.

A final paragraph in the introduction is needed to summarize the paper structure.

Also, a Literature review section is missing. This is critical in order to give importance to the paper research gap. What about research questions? I can not find them.

Paper research gap: This part is very general and lacked alignment to the research findings, no discussion was provided to derive the implication. Theoretical and pragmatics implication are vague and need to be better aligned with this paper theoretical underpinnings. In addition, the author should make references to what is the originality and value of the research work to the industry. Why is your approach necessary to compare search engine optimization results in the academic environment? Explain in detail.

There is no Conclusions part. In this new part, the authors should include a final paragraph describing the “Social implications" and "originality or value" of this study. What practical/professional and academic consequences will it have for the future? In addition, authors should make references to what is the originality and value of the research work to the industry. What practical/professional and academic consequences will it have for the future?

What about limitations? Please explain in detail.

The authors should follow this structure:

1. Introduction (ok)
2. Literature Review/Related Work (missing) -research questions should be placed here)
4. Methodology (and sub-methodology parts such as Sample…)
5. Analysis of results
6. Discussion (re-write)
7. Conclusion (sub-conclusions parts) (missing)
8.1 Managerial Implication (missing)
8.2. Practical/Social Implications (missing)
8.3. Limitations (missing)

I would also urge the author to read the articles listed below before completing the manuscript revision, therefore, cite them. The author will understand that the article structure can be improved as well as the research question justifications and literature review section:

Saura, J.R. and Bennett, D. (2019). A Three-Stage Methodological Process of Data Text Mining: A UGC Business Intelligence Analysis. Symmetry-Basel. doi: 10.13140/RG.2.2.11093.06880

Zhang, S., & Cabage, N. (2017). Search engine optimization: comparison of link building and social sharing. Journal of Computer Information Systems, 57(2), 148-159.

Berman, R., & Katona, Z. (2013). The role of search engine optimization in search marketing. Marketing Science, 32(4), 644-651.

Round 2

Reviewer 3 Report

The authors have not made the requested changes indicated by this reviewer completely. Although some changes have been made, there are still significant weaknesses that should be added before consideration. This reviewer considers that this paper, due to its originality, can receive many cites if it is accepted, but changes should still be made as indicated.

For example, the authors stated in their responses to the reviewer that it is important to understand this added paragraph: “However, it should be stressed that the relevance ranking algorithm of academic search engines differs from that of standard search engines. The ranking factors employed by the respective search engine types are not the same and, therefore, many of those used by SEO are not applicable to ASEO while some are specific to ASEO.

However: What are the main indicators of ASEO that exist so far in the literature?  Who has discovered them?

Authors should make a table highlighting these factors and the authors who discovered them or demonstrated their influence on ASEO.

The Literature Review section is weak. In addition, research questions should be moved formally at the end of the introduction or in an independent section, but not in the literature section. The reader of the article should understand the research questions in the introduction and check their viability in the literature section.

Research questions: The authors should also cite other authors who proposed similar research questions to justify the proposed.

The authors stated that the “do not consider” that a paragraph should be added at the end of the introduction indicating the structure of the article. This reviewer does. It is a standard format in academic writing.

E.g. “The remainder of this manuscript is as follows. First, we present… […] ”.

Other:

Applied to the two most commonly used academic search engines (i.e. Google Scholar and Microsoft Academic)”. Who did state that these two are the most commonly use search engines in academia? Please cite the research study or statistical report. There are others such as Sciencedirect, Scilit… etc.

"However, in the case of Google, more than 200 factors intervene." Please cite the author who stated that.

"25,000 searches were conducted." Why 25,000? Why did you choose that sample? Please cite authors.

"With both one and two-term keywords". Why did you not use long-tail keywords? Long tail keywords are the most used keywords in SEO as stated by Matt Cutts, Rand Fishkin or Avinash Kaushik. Explain in detail.

Although the Figures supposedly explain the results, the authors do not analyze them in depth or explain them in detail. This should be changed.

Round 3

Reviewer 3 Report

The authors have made the indicated changes. This reviewer does not have additional comments.